# Risk Factors Analysis of Surgical Infection Using Artificial Intelligence: A Single Center Study

**DOI:** 10.3390/ijerph191610021

**Published:** 2022-08-14

**Authors:** Arianna Scala, Ilaria Loperto, Maria Triassi, Giovanni Improta

**Affiliations:** 1Department of Public Health, University of Naples “Federico II”, 80100 Naples, Italy; 2Interdepartmental Center for Research in Health Care Management and Innovation in Health Care (CIRMIS), University of Naples “Federico II”, 80100 Naples, Italy

**Keywords:** health informatics, statistical software, sursgical site infection

## Abstract

**Background:** Surgical site infections (SSIs) have a major role in the evolution of medical care. Despite centuries of medical progress, the management of surgical infection remains a pressing concern. Nowadays, the SSIs continue to be an important factor able to increase the hospitalization duration, cost, and risk of death, in fact, the SSIs are a leading cause of morbidity and mortality in modern health care. **Methods:** A study based on statistical test and logistic regression for unveiling the association between SSIs and different risk factors was carried out. Successively, a predictive analysis of SSIs on the basis of risk factors was performed. **Results:** The obtained data demonstrated that the level of surgery contamination impacts significantly on the infection rate. In addition, data also reveals that the length of postoperative hospital stay increases the rate of surgical infections. Finally, the postoperative length of stay, surgery department and the antibiotic prophylaxis with 2 or more antibiotics are a significant predictor for the development of infection. **Conclusions:** The data report that the type of surgery department and antibiotic prophylaxis there are a statistically significant predictor of SSIs. Moreover, KNN model better handle the imbalanced dataset (48 infected and 3983 healthy), observing highest accuracy value.

## 1. Introduction

Healthcare associated infections (HAI) are among the major complications of modern medical therapy [1]. The most important HAIs are those related to invasive devices: central line-associated bloodstream infections (CLABSI), catheter-associated urinary tract infections (CAUTI), ventilator-associated pneumonia (VAP) as well as surgical site infections (SSI). Surgical site infections (SSIs) are infections that arise after surgery and can affect the skin, an organ and part of the body [2]. The last ECDC Report for SSIs shows that in 2017, 10.149 SSIs were reported using patient-based or unit-based surveillance. Of these 47% were superficial, 30% deep and 22% organ/space SSIs. Thirty-four per cent of the SSIs were diagnosed in hospitals, whereas 52% were detected after discharge [3]. The reported prevalence ranges from 0.5% to 10.1% [3]. The latest Italian HAI prevalence study reports a prevalence of 8.03%. Of these infections, 16% were SSIs with a prevalence of 1.27% [4].

Most of the surgical site infections (SSIs) arise in secondary care and are often associated with antibiotic-resistant organisms, such as methicillin-resistant Staphylococcus aureus. A long list of potential patient’s risk factors for SSI has been identified, but few have been confirmed as such in randomized clinical trials. SSIs represent about a fifth of all healthcare-associated infections [5]. The SSIs cost usually is defined as hospital charges for additional goods and resources. In patients with SSI, costs are divided into direct and indirect. Direct costs include prolonged hospitalization and readmission to the hospital, outpatient visits, visits to the emergency department, additional surgery, and prolonged antibiotic therapy, while the indirect costs are difficult to quantify because they include lost productivity not only by the patient but also by family members or friends. For this reason, the true cost of an SSI often is unknown [6]. There are multiple factors contributing to HAI and including healthcare associated factors, environmental factors, and patient-related factors. Healthcare factors include the use of invasive devices, the type of surgery surgical procedures and pressure from excessive antibiotic use as prophylaxis. The environmental factors include contaminated air-conditioning systems and the physical layout of the facility (e.g., open units with beds close together). Patient’s related factors include genetic factors, the severity of underlying illness, use of immunosuppressive agents and prolonged hospital stays.

It is recognized now that many SSIs are partially preventable and that healthcare can become safer. A passive strategy can be used for SSI reduction in which surveillance protocols lead to infection reduction through timely and timely feedback [7]. The classic study that demonstrated the importance of this was the SENIC study, funded by the United States Centers for Disease Control and Prevention (CDC) and included 338 randomly selected hospitals stratified by geography, bed capacity and teaching status. The work showed that infection control programs with dedicated hospital epidemiologists and surveillance programs reduced nosocomial infections by 32% compared to facilities without infection control programs [8]. To design an effective prevention program, it is necessary to consider the impact that SSIs have on the length of hospital stay [9,10,11,12], which is a performance indicator of the quality of health processes [13,14,15,16,17,18,19]. In addition, identifying risk factors associated with SSIs can help reduce the incidence of SSIs [11,20] and add value to HTA studies, which are widely used to support health decision-making [21,22,23,24,25]. This paper presents a logistic regression model to study the impact that different clinical, demographic and organizational factors have on the risk of occurrence of SSIs in a surgery department and an AI model to predict the risk of infection has been used.

## 2. Materials and Methods

The study population include all patients that underwent surgery at the surgical departments of the “Federico II” University Hospital in Naples (Italy) between 2015 and 2019. Active, patient-based surveillance for surgical site infection (SSIs) is continually performed by trained healthcare staff in all surgery departments of the Hospital. The Protocol adopted in the Campania Region for the surveillance of SSIs corresponds to that of the SNICh national surveillance system [26,27]. It defines which interventions to monitor, how and for how long to carry out surveillance, it indicates the information to be collected for each intervention, provides definitions for each of the variables of interest, such as diagnosis of surgical site infection, class and type of intervention, duration of intervention, ASA score, risk index, etc. Data collection is carried out prospectively for all patients undergoing selected surgical interventions. All patients who meet the inclusion criteria in the chosen surveillance period (1 year) are included without any selection. Patients must be supervised, if necessary, even after discharge, for a period of 30 days after surgery in the case of surgical interventions that do not involve the placement of prostheses; the follow-up must be continued for 90 days for patients undergoing interventions with prosthetic material implantation. The ICAARO web IT platform has been implemented since 2015. All companies in the Campania Region can access the platform and manage data from the surveillance of Infections Related to Assistance. ICAARO web allows direct entry of data collected as part of surveillance activities, as well as the extraction of specific reports for each surveillance [27].

For this study, no patients’ informed consent, nor local Ethical Committee authorization was required, as all the data come from HAIs surveillance that is regulated by the Regional Health Authority as defined in the Regional Plan for Healthcare-associated infections Prevention and Control [28]. Data were collected retrospectively using ICARO web IT reports and QuaniSDO, i.e., the system employed for the computerization of hospital discharge records,. A risk folder must be completed for all patients hospitalized from the time of admission to discharge. This folder contains epidemiological data important for the study of hospital infections. All diabetic patients and patients treated with corticosteroids were excluded because these two conditions slow wound healing and predispose to bacterial infections. The information, extracted from medical records, are the following:
Gender (male/female);Age;Length of stay (days);Hospital regime;Surgery department;Number of antibiotics;SSI (yes/no).

In accordance with the definition of the National Nosocomial Infection Surveillance System (NNIS), surgical site infection (SSI) is defined as an infection that occurs within 30 days of surgery (or within 1 year if thereafter an implant is left in place during the surgical procedure, i.e., an implantable foreign body, of non-human origin) and which may involve the incisional or deep tissue at the site of the surgery [29].

All data relating to the SSIs were collected by the surgeon who performed the surgery. The surgeon also supervised the patient both during hospitalization and for a period of 30 days following discharge, as recommended by the Centers for Disease Control and Prevention (CDC) criteria [2]. Post-operative follow-up could be done during an outpatient visit or by telephone interview. In addition, upon discharge, each patient received a pre-printed questionnaire, in order to record the onset of any symptoms of HAI (Hospital Acquired Infections) during the follow-up period.

### 2.1. Statistical Analysis

Before performing the statistical tests, the distributions were analyzed using the Shapiro-Francia test. For parameter age and parameter length of stay, the test showed a non-normal distribution. For this reason, non-parametric tests were used. Specifically, Kruskal-Wallis and statistic tests have been implemented to obtain population characteristics. The Kruskal-Wallis test is a rank-based nonparametric test that can be used to determine if there are statistically significant differences between two or more groups of an independent variable on a continuous or ordinal dependent variable. A *p*-value below the threshold of 0.05 was considered significant for the above tests [30,31,32,33,34,35].

Logistic regressions were used to test the association between the SSIs (as dependent variable) and the different risk factors under study (as explanatory variables). The explanatory variables are: sex, age, hospital regime, surgery department, length of preoperative and postoperative hospital stay and antibiotic prophylaxis. The multivariate model was adjusted on the risk factors considered. Associations were deemed significant if *p*-values were below the threshold of 0.05. Sensitivity analyses were performed using Firth’s penalized maximum likelihood logistic regression. Data were analyzed using STATA version 15.

### 2.2. Predictive Analysis

Machine learning algorithms are learning functions that allow mapping input variables to an output value with the aim of making predictions. This initial learning task allows to subsequently classify given new samples of the same input variables. Different Artificial Intelligence models are used: Random Forest (RF), Logistic Regression (LR), Decision Tree (DT), K-Nearest Neighbors (KNN), Gradient Boosted Tree (GBT), XGBoost (XGB) and Naive Bayes (NB). The target value (SSI) is influenced by the input variables. Starting from the knowledge acquired through the analysis of the initial set of data called training, the model was built. For this reason, the dataset was divided into training (75%) and test (25%) sets. Since the dataset was unbalanced in terms of the presence of infections, the Synthetic Minority Over-sampling Technique (SMOTE) was used [36]. Some supervised learning algorithms (such as decision trees and neural networks) require an equal distribution of classes to generalize well, i.e., to achieve good classification performance. In case of unbalanced input data, e.g., there are only a few objects of the “active” class but many of the “inactive” class, this node adjusts the distribution of classes by adding artificial rows (in the example adding rows for the “active” class).

The algorithm works approximately as follows: It creates synthetic rows by extrapolating between a real object of a given class (in the above example “active”) and one of its nearest neighbors (of the same class). It then chooses a point along the line between these two objects and determines the attributes (cell values) of the new object based on this randomly chosen point.

## 3. Results

### 3.1. Statistical Analysis

In this section, we report the performed statistical analysis in order to investigate the correlation between different factors (independent variables) and the risk of infection (dependent variable). Data include a total of 4031 patients that underwent surgery at the surgery department of the “Federico II” University Hospital in Naples (Italy). Patients affected by SSIs underwent surgery in ordinary hospitalization, precisely, in maxillofacial or orthopedic surgery, had longer preoperative and postoperative LOS, and were undergoing therapy with 2 or more antibiotics (Table 1).

This section may be divided into subheadings. It should provide a concise and precise description of the experimental results, their interpretation, as well as the experimental conclusions that can be drawn.

Table 1 shows that except for sex, all variables chosen to conduct this study are statistically significant.

The Kruskal-Wallis test revealed that there is a statistically significant difference between the different groups of the independent variable. The significance level is *p* = 0.0026 for age and *p* = 0.001 for LOS, which are below 0.05. Multivariate analyses confirmed that maxillofacial, orthopedics and antibiotic prophylaxis with 2 or more antibiotics were significant predictors of suffering from SSIs (Table 2).

### 3.2. Predictive Analysis

In this section, a predictive analysis was performed based on several risk factors that were initially normalized and then processed to predict whether or not a patient had SSIs. The results obtained from the predictive analysis were evaluated in terms of accuracy, precision, sensitivity, specificity and F-measure.
$$Accuracy=\frac{Number\;of\;correct\;predictions}{Total\;number\;of\;predictions}$$
$$-measure=2*\frac{\left(precision*recall\right)}{precision+recall}$$

The performances of selected ML algorithms were analyzed. The table shows the results obtained.

The best performance has been obtained with RF, DT and KNN (Table 3).

In this case, the best algorithm selected has been KNN for its simple structure compared with RF and DT. The confusion matrix of these tree algorithms has been reported in Table 4, Table 5 and Table 6.

### 3.3. Global Feature Importance

Feature importance is calculated by counting how many times it has been selected for a split and at which rank (level) among all available features (candidates) in the trees of the random forest. A higher value indicates higher feature importance.

Figure 1 shows that the hospitalization regime is the most important feature of all, followed by preoperative LOS and department. Sex seems to be the characteristic that affects infections the least.

## 4. Discussion

The factors that most influence the risk of SSIs were evaluated in order to study the unbalanced datasets, where the size of the class “infected patients” was lower than that of the “un-infected patients” and for this reason the “Firth’s penalized maximum likelihood logistic regression” was used, in fact, this type of regression is well-suited for unbalanced datasets and for the regression analysis of rare events. The obtained results demonstrated how the level of contamination of the type of surgery can impact significantly the infection rate (*p*-value < 0.005) since it is obvious that Maxillo-facial and Orthopedics surgery has a higher risk of provoking infections. In the scientific literature, in fact, several studies report that surgical procedures in the Maxillo-Facial area are at high risk of infection. For example, Cunha et al. [37] discuss the risk of infection for head and neck oncological surgery, while Cousin et al. [38] for orthognathic surgery.

In addition, the length of preoperative hospital stay cannot increase the rate of surgical infections (*p*-value = 0.071). In fact, it is known that the length of post-operative stay and the risk of infection are correlated. Mujagic et al. [12] show that although there is no significant independent association between preoperative length of stay and risk of SSI, postoperative LOS were significantly associated with SSI. The latter is also confirmed by the various studies reviewed by Manoukian et al. [39]. With regard to orthopaedic surgery, there is evidence of an increase in infection levels due to Enterobacteriaceae, together with high use of hip and knee replacement surgery and an increasingly obese surgical population, for which there is a higher risk [40].

Furthermore, data revealed that antibiotic prophylaxis is a possible determining factor, particularly, the prophylaxis with 2 or more antibiotics is the most significant predictor of the model (*p*-value = 0.004). This is likely to be related to the characteristics of patients who are prescribed combined antibiotic therapies and who are more fragile. This result is in line with what Nasuzione et al. [41] reported on patients with chronic kidney disease. Hawn et al. [42], on the other hand, show that the risk of SSI varies with patient and procedure factors, as well as antibiotic properties, but is not significantly associated with the prophylactic timing of the antibiotic.

In general, the study of risk factors for SSIs has been analyzed in several articles. Fisichella et al. [43] show that for orthopedic surgery there is a correlation with age, contrary to our study, and with diabetes and smoking. Among these, diabetes is then also significantly correlated with SSI risk for cardiothoracic surgery, as shown by Latham et al. [44].

In sum, the goal of the work is to increase the knowledge of health professionals on such an important topic as SSIs. In particular, classic statistical analysis is combined with the construction of predictive algorithms to determine whether or not a patient is suffering from an infection. Although the use of Machine Learning in this field is not new, e.g., Montella et al. [45] have already used it to study healthcare-associated bloodstream infection in neonatal intentional care or Tunthanathip et al. [46] specifically for neurosurgical operation, in our work we analyze the situation of the whole hospital by including both organizational and clinical-demographic factors of the patients. Knowing a priori through these algorithms the risk of SSI could have a direct impact on the care practices implemented by the hospital.

The limitations of our work are several. One of the main limitations of the study is the retrospectiveness of the analysis. In fact, although the surveillance is active and carried out in real-time, the data analysis is carried out retrospectively together with the data deriving from the hospital discharge forms. It would be desirable to be able to analyze the data in parallel with their collection to create an active, flexible and responsive surveillance system. Another limitation is related to the small period of observation and variables included in the model. As discussed earlier, several additional clinical factors, such as diabetes [42,43], have been shown to be significantly correlated to SSI risk but were not included in this work. In addition, further mathematical tools may be implemented on the new dataset [46]. Finally, processes to identify causes and solutions of the most critical cases were not analyzed. Approaches such as Lean Six Sigma [47,48,49] or Fuzzy Logic [50] have been shown to be valid supports for reducing the risk of infection and will, therefore, be the subject of future studies.

## 5. Conclusions

The risk factors correlated to the surgical site infection that occurs in patients that underwent surgery were evaluated in this work. The court of patients of the “Federico II” University Hospital is composed of 4031 patients (48 with SSIs). A performed statistical analysis to correlate SSIs and factor risk was carried out, in fact, Firth’s penalized maximum likelihood logistic regression model allowed us to investigate the factors most influencing the risk of infection. The data relived have noted that there is a statistically significant difference between SSIs and the type of surgery department and between SSIs and antibiotic prophylaxis, if the antibiotic number is 2 or more. Finally, different artificial intelligence models to predict SSIs were used. In this analysis, it is easy to note how KNN better handle the imbalanced dataset (48 infected and 3983 healthy), observing the highest accuracy value. In further work, we will aim to expand the dataset and widen the number of predictors in order to build a more reliable and accurate model.

## Figures and Tables

**Figure 1 ijerph-19-10021-f001:**
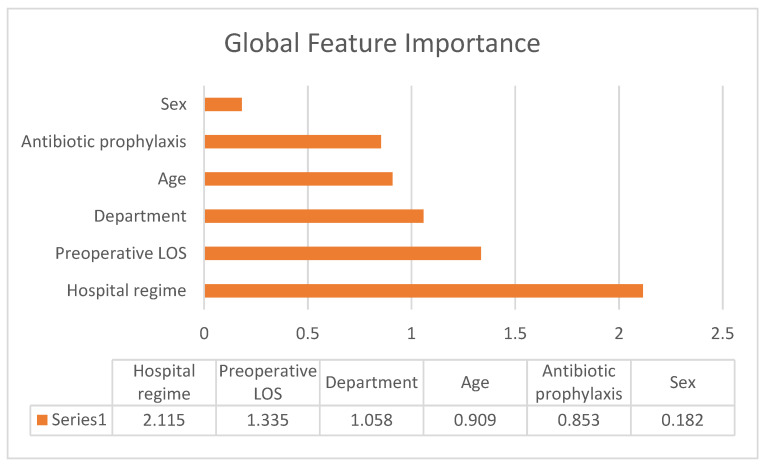
Global Feature Importance.

**Table 1 ijerph-19-10021-t001:** Study population characteristics.

	SSIs	Non-SSIs	*p*-Value
Sex, boys	29 (1.23%)	2331 (98.77%)	0.791
Sex, girls	19 (1.14%)	1652 (98.86%)	
**Hospital Regime**			
Ordinary hospitalization	48 (1.33%)	35,558 (98.67%)	0.017
Day Surgery	0 (0.00%)	425 (100.00%)	
**Surgery Department**			
General surgery	2 (0.25%)	796 (99.75%)	<0.000
Maxillo-facial surgery	36 (3.17%)	1098 (96.83%)	
Pediatric surgery	1 (0.12%)	813 (99.88%)	
Neurosurgery	4 (0.42%)	958 (99.58%)	
Orthopedics	5 (1.55%)	318 (98.45%)	
**Antibiotic prophylaxis**			
no	6 (0.57%)	1.038 (99.43%)	
1 antibiotic	19 (0.70%)	2697 (99.30%)	<0.000
2 or more antibiotics	23 (8.49%)	248 (91.51%)	

**Table 2 ijerph-19-10021-t002:** Associations between SSIs and the risk factors in surgery patients.

	OR	95% CI	*p*-Value
**Sex, girls**	0.750	−0.705–6.228	0.349
**Age**	1.010	−0.006–0.026	0.221
**Hospital regime**			
Ordinary hospitalization	3.908	−0.705–6.228	0.248
**Surgery Department**			
Maxillo-facial surgery	7.321	0.867–3.595	<0.001
Pediatric surgery	0.660	−3.002–1.883	0.722
Neurosurgery	1.422	−1.177–2.119	0.656
Orthopedics	4.781	0.015–3.370	0.047
**Length of preoperative hospital stay**	1.021	−0.002–0.046	0.071
**Antibiotic prophylaxis, No**			
1 antibiotics	0.447	−1.797–0.318	0.151
2 or more antibiotics	4.294	0.438–2.627	0.004

**Table 3 ijerph-19-10021-t003:** ML Algorithms.

Performance Metrics	Class	RF	LR	DT	KNN	GBT	XGB	NB
**Accuracy (%)**	Overall	**92.5**	78.5	**93.9**	**94.9**	90	69.5	72.4
**Error (%)**	Overall	7.5	21.5	6.1	5.1	10	30.5	27.6
**Precision (%)**	0	99.4	99.2	99.5	99	99.2	99.7	99.4
	1	7.9	2.8	11.1	4.7	5.1	3.2	2.8
**Sensitivity (%)**	0	93	78.8	58.3	95.9	90.6	69.4	72.5
	1	50	50	94.4	16.7	41.7	83.3	66.7
**Specificity (%)**	0	50	50	58.3	16.7	41.7	83.3	66.7
	1	93	78.8	94.4	95.9	90.6	69.4	72.5
**F-measure (%)**	0	96.1	87.9	96.9	97.4	94.7	81.8	83.9
	1	13.6	5.2	18.7	7.3	9	6.1	5.4

**Table 4 ijerph-19-10021-t004:** Confusion Matrix (DT).

Real/Predicted	0	1
**0**	940	56
**1**	5	7

**Table 5 ijerph-19-10021-t005:** Confusion Matrix (RF).

Real/Predicted	0	1
**0**	926	70
**1**	6	6

**Table 6 ijerph-19-10021-t006:** Confusion Matrix (KNN).

Real/Predicted	0	1
**0**	955	41
**1**	10	2

## Data Availability

The datasets generated and/or analyzed during the current study are not publicly available for privacy reasons but are available from the corresponding author on reasonable request.

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
