# Peer review of "Risk Factors Analysis of Surgical Infection Using Artificial Intelligence: A Single Center Study"

_ijerph, 2022, doi:10.3390/ijerph191610021_

Round 1

Reviewer 1 Report

Although Authors have revised the manuscript, I think that many important issues have not been addressed. Particularly, I suggest that the paper should clearly report protocol and definitions used in the surveillance of SSIs. Limits of the study have not been addressed appropriately. One of the main limits of the study is the retrospective nature of the SSI surveillance. English language should be better revised. The manuscript lacks of recent data on the epidemiology of SSIs at European and National level.

Reviewer 2 Report

No further comments. Authors properly addressed reviewers comments

Author Response

The authors have subjected the entire text to a thorough revision

Reviewer 3 Report

All comments have been addressed by the authors

Author Response

(The authors gave the same response as above.)

Round 2

Reviewer 1 Report

The main limitation of the study is the retrospective design of the surveillance. This topic could seriously compromise the quality of the analyzed data and the obtained results. A validation study should be performed. The manuscript lacks of recent data on the epidemiology of SSIs at National level. 

Author Response

This manuscript is a resubmission of an earlier submission. The following is a list of the peer review reports and author responses from that submission.

Round 1

Reviewer 1 Report

Authors analysed surgical infection using artificial intelligence. Although idea is interesting, execution and presentation of the study should be greatly improved in order for this paper to be publishable.

Major limitations (excluding numerous minor concerns):

  • Materials and Methods section should be expanded and explained in details regarding sampling, methodology and patient inclusion
  • Tables and figures in Result section are confusing and lacking in quality, while a lot of tables are consisted of only few variables and are unnecessary
  • Discussion section is extremely limited, and as such cannot be relevant for this topic. Discussion should be extensively expanded. 

Reviewer 2 Report

The aim of the study was to investigate risk factors of SSIs. Topic should be interesting, particularly for the AI analysis even if results obtained should be properly revised. 

  • Lenght of postoperative hospital stay is strictly related to SSIs but it is not a risk factor, somewhat an effect.
  • Why patients underwent 2 or more antibiotics? This is not reported in any guidelines...
  • The AI analysi is cahotic and should be revised. 
  • Reading the title, readers expect to obtain suggestion on reducing SSI, however results obtained avoid this

Reviewer 3 Report

The aim of the study is interesting and the addressed topic very important.

However, in order to improve the manuscript, I suggest the following amendments.

The text should be reviewed to improve English language. Furthermore, the Authors should pay attention to standardized the terminology, for example, “antibiotic preventive treatment” “

 antimicrobial therapy” and “antibiotic prophylaxis”.

In the conclusion section of the abstract the authors mention for the first time “…there is a statistically significant difference between SSIs and the type of surgery department”, I suggest to rewrite this section.

The Introduction section should be focused on SSI: the first part is not usefully to address the aim of the study.

References should be updated in order to give the most recent data on the epidemiology of SSIs also considering European and National data.

The period of the study should be clearly reported.

Results should be better organized. Moreover, please provide a legend for the Figure 1.

I suggest to start the discussion section with a brief summary of the main findings of the study. Limitations of the research should be addressed and discussed.